Non-destructive monitoring of 3D cell cultures: new technologies and applications

Cortesi Marilisa marilisa.cortesi2@unibo.it 1 2
Giordano Emanuele 1 3 4
1 Department of Electrical, Electronic and Information Engineering ”G.Marconi”, University of Bologna , Bologna , Italy
2 School of Clinical Medicine, Faculty of Medicine and Health, University of New South Wales , Kensington , Australia
3 BioEngLab, Health Science and Technology, Interdepartmental Center for Industrial Research (HST-CIRI), University of Bologna , Ozzano Emilia , Italy
4 Advanced Research Center on Electronic Systems (ARCES), University of Bologna , Bologna , Italy
Gillespie Joseph
Electronic publication date: 2022 May 12
Publication date: 2022
Volume: 10
Electronic Location ID: e13338
Received 2022 Jan 12; Accepted 2022 Apr 5
Copyright: ©2022 Cortesi and Giordano
Copyright year: 2022
Copyright holder: Cortesi and Giordano
License: This is an open access article distributed under the terms of the Creative Commons Attribution License, which permits unrestricted use, distribution, reproduction and adaptation in any medium and for any purpose provided that it is properly attributed. For attribution, the original author(s), title, publication source (PeerJ) and either DOI or URL of the article must be cited.
License URL: https://creativecommons.org/licenses/by/4.0/

Keywords: 3D cell culture, Non-destructive technology, in-vitro quantitative analysis

Funding: POR-FESR (2014-2020) Emilia Romagna (PG/2018/632022 DINAMICA) Italian Ministry for Education, University and Research (MIUR) under the program “Dipartimenti di Eccellenza (2018-2022)” Marie Sklodowska-Curie 883172 This article is the result of the research project funded by POR-FESR (2014-2020) Emilia Romagna (PG/2018/632022 DINAMICA). This work was supported by the Italian Ministry for Education, University and Research (MIUR) under the program “Dipartimenti di Eccellenza (2018-2022)”. This project received funding from the European Union’s Horizon 2020 research and innovation programme under the Marie Sklodowska-Curie grant agreement No 883172”. There was no additional external funding received for this study. The funders had no role in study design, data collection and analysis, decision to publish, or preparation of the manuscript.

==============================
3D cell cultures are becoming the new standard for cell-based in vitro research, due to their higher transferrability toward in vivo biology. The lack of established techniques for the non-destructive quantification of relevant variables, however, constitutes a major barrier to the adoption of these technologies, as it increases the resources needed for the experimentation and reduces its accuracy. In this review, we aim at addressing this limitation by providing an overview of different non-destructive approaches for the evaluation of biological features commonly quantified in a number of studies and applications. In this regard, we will cover cell viability, gene expression, population distribution, cell morphology and interactions between the cells and the environment. This analysis is expected to promote the use of the showcased technologies, together with the further development of these and other monitoring methods for 3D cell cultures. Overall, an extensive technology shift is required, in order for monolayer cultures to be superseded, but the potential benefit derived from an increased accuracy of in vitro studies, justifies the effort and the investment.

Introduction

Culturing cells in vitro is a cornerstone of biomedical research, which has been instrumental for furthering our scientific knowledge and has enabled the more ethical development of safe and effective drugs. Indeed, the possibility of maintaining cells in the laboratory has provided researchers with a platform for the study of physiological and pathological phenomena which was effective and inexpensive, and afforded them a higher level of control with respect to entire organisms.

In recent years, however, relevant differences between in vitro and in vivo behavior have emerged (Breslin & ODriscoll, 2013; Waring et al., 2015). These are generally attributed to the over simplification of 2D cell monolayers, which are unable to capture relevant features of in vivo biology and consequently results in poor translation of laboratory data in clinical studies (Honek, 2017; Liston & Davis, 2017; Menshykau, 2017; Jo et al., 2018).

The main limitation of 2D cell cultures is that the microenvironment where the cells are maintained completely disregards the contribution of the physical cues associated with the physiological extracellular matrix (ECM) on cell behaviour. This limits cell-to-cell signalling and impacts over their functions. For example Edmondson et al. (2014) showed that 2D cell culture can affect the expression of cell surface receptors which, in turn, could alter the response to drugs targeting these receptors (Attwood et al., 2020; Abbas et al., 2021). Furthermore, a comparison of 2D vs 3D cell cultures led to the identification of important differences in proliferation and treatment response (Li et al., 2021b) .

A partial solution to these issues involves substituting the flat polystyrene plates with micro or nano-patterned surfaces. While different techniques and materials can be used to produce the grooved pattern, and specific motifs tend to be better suited for different applications, this strategy has been shown to be effective in (i) recapitulating cell response to multi-cue environments, (ii) enhancing adhesion, proliferation, migration and stem-cell differentiation, and (iii) enable large scale cell alignment (Table 1).

Table 1 Micro and nano patterning effect on cell cultures.

Application	References	
Response to multi-cue environments	Van Der Putten et al. (2021), Chang & Taniguchi (2021)	
Adhesion, proliferation, migration and	Han et al. (2021), Kang et al. (2021), Wang et al. (2021b)	
Differentiation enhancement	Blin (2021), Nagayama & Hanzawa (2022)	
Large-scale cell alignment	Yu et al. (2021)	

3D cell culture models represent a more comprehensive solution. These setups integrate additional features of biological tissues (e.g., tridimensional cell disposition, interaction among cells and with the surrogate extracellular matrix, non-uniform nutrients availability) and as such have been shown to reduce the gap between in vitro and in vivo behavior (Duval et al., 2017; Kapałczyńska et al., 2018; Nunes et al., 2019; Costard et al., 2021; Belfiore et al., 2021).

The widespread use of these setups and their integration in the drug development pipeline are however hampered by the techniques used for their analysis. Indeed, methods that require the fixation or disgregation of the culture can be generally considered a standard de facto. This inability to monitor 3D cell cultures over time affects the accuracy of the results, as small differences in initial cell number or structure size cannot be accounted for. Additionally, a larger number of constructs will be needed, hence increasing the resources necessary to conduct the experiments.

In an effort to overcome this adoption barrier, in this review paper, we showcase non-destructive techniques for the analysis and characterization of relevant features in 3D cell cultures. Each section analyzes a specific kind of assay and a comparative analysis of non-destructive and destructive techniques will be presented, together with its perspective evolution in the near future. Additionally, an introductory paragraph, aimed at describing the main 3D cell culture approaches currently available, and a concluding section summarizing the status of non-destructive analysis of 3D cell cultures have been included.

3D Cell Culture Models

While a comprehensive analysis of 3D cell culture methods is beyond the scope this review, in this section we provide an overview of the most commonly available methods, to lay the foundation for the understanding of this work. For a more comprehensive description of 3D cell culture technologies, the reader is referred to other recently published review papers such as (Lv et al., 2017; Badekila, Kini & Jaiswal, 2021; Duval et al., 2017).

3D cell cultures are classically divided between scaffold-based (SB), that is relying on an external support material, and scaffold-free (SF), where cells self-assemble in clusters or spheroids (Fig. 1). Within these groups, multiple practical protocols are available.

Figure 1 Summary of 3D cell culture models.

In (A) the scaffold-based (SB) setups are presented, while (B) and (C) illustrate scaffold free (SF) and hybrid methods, respectively.

For SB systems, the scaffolds can either be produced independently from cell seeding (Liverani et al., 2019; Dinescu et al., 2019) or aggregated with cells already included (Dollinger et al., 2017; Pasini et al., 2021). The former generally grants more flexibility in scaffold production, allowing for a wider range of materials and technologies to be used. Non-vital techniques, such as freeze-drying, two- photons polymerization, and electrospinning enable the fine-tuning of the scaffold’s mechanical properties and superficial features (Remuzzi et al., 2020; Dinescu et al., 2019; Wang et al., 2021a), which results in structures that are stable over time and well suited to withstand mechanical stimulation (Lovecchio et al., 2019).

Including cells in the scaffold’s liquid phase tends to be associated with their more uniform distribution throughout the structure, as they can be dispersed within the solution, rather than having to migrate from the surface. This feature limits the possible scaffold materials to polymers that become solid through temperature changes or gentle chemical processes that do not compromise cell viability. The resulting structure, also called hydrogel, is often less structurally robust and more prone to macroscopic changes in size due to cell activity (James-Bhasin, Siegel & Nazhat, 2018).

A fundamental advancement in scaffold production is represented by bioplotting, which relies on fused deposition modeling (FDM) technology to print 3D structures with high precision and fidelity (Buenzli et al., 2020). In most cases, cells can be directly dispersed within the ink, thus combining high accuracy in scaffold production techniques with uniform cell distribution (Ahlfeld et al., 2020).

SF set-ups, on the other hand, rely on cells’ self-aggregation/organization properties to produce approximately spherical 3D cultures. These are generally smaller than the ones obtained with SB approaches, with starting populations of about 10 k cells (Shoval et al., 2017; Gamerith et al., 2017; Ahmad et al., 2017; Skeberdyté et al., 2018) as opposed to 100 k cells (Pasini et al., 2021; Picone et al., 2020; Rivero et al., 2020), but tend to be simple and, in some cases, well suited for high throughput analyses (Benien & Swami, 2014). The reduced size of SF cultures has however been shown to be an important drawback of these systems, as the number of cells that they can support is very low, when compared with those that compose human tissues (De Pieri, Rochev & Zeugolis, 2021). Furthermore, the lack of independent support structure results in densely packed cells which can be associated with the presence of hypoxic cores and macromolecular crowding (De Pieri, Rochev & Zeugolis, 2021), that further reduce the accuracy of these systems.

The ease of production of SF 3D cell cultures, which relies on various methods to prevent or disrupt cell adhesion to the culturing surface, however, makes them an attractive setup. Low attachment plates are widely used in this regard (Takagi et al., 2007), as the hydrophilic polymer coating on their surface prevents cells from attaching and thus enables spheroid formation. This approach is simple, and generally leads to spheroid formation within a few hours, but provides little control on the size of each culture and thus tends to be associated with higher variability. To address this limitation, culture surfaces micro-patterned with microscopic valleys of defined shape and size can be used. In this case, cells are generally allowed to adhere to a micro-patterned thermosensitive substrate, to ensure an approximately uniform cell density within each valley. The temperature of the structure is then changed, to induce a shape change in the substrate which, in turn, leads to cell detachment and spheroid formation. An example of this approach is reported in Kim et al. (2018a), where decreasing the temperature to 4 °C led to a significant increase in micropattern area which resulted in spheroid formation within minutes. Hanging drops is another widely used method in which cells are cultured within a suspended drop of medium and aggregate, due to gravity, to its bottom (Wang et al., 2017; Cho et al., 2020; Zibaei et al., 2021). This technique was developed at the beginning of the twentieth century to visualize small live organisms and study their motility (Jain, Jain & Jain, 2020) and then adapted to the production of spheroids.

The dichotomy between SB and SF methods, however, is not perfect and numerous approaches combine elements typical of both methods. Indeed, polymeric mixtures such as Matrigel can be used when producing spheroids to allow for the aggregation of more cells and increase repeatability (Badea et al., 2019). At the same time, integrating external support structures and cells’ self-aggregating properties results in organoids, 3D cultures integrating multiple cell types that aim at replicating the structure and functionality of organised tissues (Lu et al., 2019; Asai et al., 2017). Microspheres are another notable example, as they serve the double purpose of cell culture support and precision delivery system (Dong et al., 2021). Many different techniques are available for their production (Zhang et al., 2021), and some of them allow for the inclusion of living cells which have been shown to be able to survive, proliferate and differentiate within these setups (Zhang et al., 2018).

This wealth of options for 3D cultures effectively allows to choose the best set-up for each application, as different models are best suited to replicate specific aspects of the corresponding in vivo system (Kapałczyńska et al., 2018). At the same time, the lack of a single established substitute for 2D cultures has contributed to delaying the introduction of 3D cultures in the drug development pipeline (Kelm et al., 2019).

This work, and the experimental techniques it showcases, are expected to contribute to the translational value of 3D cell cultures and their general applicability in biomedical research.

Cell Viability and Proliferation

The evaluation of cell viability is key to a number of in vitro studies. Indeed, it is used to test the feasibility of new cell culture methods (Yu et al., 2020; Paez et al., 2020), assess the efficacy of potential therapeutic treatments (Marrella et al., 2021; Yang et al., 2020), and generally monitor the health and status of a population of cells (Vitale et al., 2020; Rivero et al., 2020).

The most common approaches for the evaluation of cell viability are metabolic membrane structural competence and DNA-content based assays which repeated over time allow for the quantification of proliferation (Table 2).

Table 2 Cell viability and proliferation assays.

Assay	References	
Metabolic	Skeberdyte et al. (2020), Rivero et al. (2020), Hercog et al. (2020)	
	Fontoura et al. (2020), Huang, Yu & Tang (2020)	
Membrane structural competence	Vitale et al. (2020), Paez et al. (2020), Khan et al. (2020)	
	Hilderbrand et al. (2020), Ergene et al. (2020), Buenzli et al. (2020)	
	Ahlfeld et al. (2020), Marrella et al. (2019), Yang et al. (2018)	
	Pepelanova et al. (2018), Raphael et al. (2017), Ahlfeld et al. (2017)	
DNA content	Zargarzadeh et al. (2022), Eswaramoorthy et al. (2021), Santos, Custódio & Mano (2022)	
BrdU assay	Racané et al. (2021), Alkildani, Jung & Barbeck (2021), Sevimli et al. (2022)	

Bromodeoxyuridine (BrdU) assay is another established way of quantifying cell proliferation, through the visualisation of newly synthetised DNA in proliferating cells (Table 2).

Metabolic methods are a class of assays in which an optical (absorbance or fluorescence) signal, proportional to the number of living cells, is generated through a chemical modification of a soluble reagent added to the culture. These methods were initially developed for 2D cultures and, depending on the reagent’s composition, can preserve cell viability. One of the main drawbacks of translating this approach to 3D cell cultures is that it requires the reagent to diffuse through the whole structure. This condition might not be verified, especially for larger SB cultures, thus the results might be partial or inaccurate. For this reason, live-dead staining tends to be preferred. This method consists in staining a thin slice of the culture with both calcein-AM and ethidium homodimer-1. These compounds produce strong fluorescent signals, at different wavelengths, when reacting with alive or dead cells respectively and can thus be used to quantify their density.

This technique has the advantage of maintaining cell spatial distribution, and thus allows to study how different positions within the structure affect cell behaviour. On the other hand, it does not preserve cell viability and requires the sectioning of the culture, thus making this method poorly suited for the long term monitoring or high throughput experiments.

Another class of non-viable methods for viability quantification relies on the evaluation of DNA content using fluorescent probes. These assays are well established, accurate, and can be used for 3D cell cultures with minimal adjustment with respect to 2D monolayers, but they require the DNA to be extracted, thus resulting in the impossibility of monitoring the same culture over time. Additionally, this is an indirect measure, that determines the number of cells as the ratio of total amount of DNA in the sample and the assumed quantity of DNA present within each cell. As the latter can vary widely between different cells (Gillooly, Hein & Damiani, 2015) and with the different phases of the cell cycle (Cooper, 2000), this technique could result in non-negligible errors, especially when comparing samples where cells have different sizes or where the length of the cell cycle might be affected.

BrdU assay directly quantifies proliferation through a thymidine analog that, when added to the culture, is incorporated in the DNA of proliferating cells. The quantification of the BrdU, however, is based on immunostaining, which requires cells to be fixed and permeabilized, in order for antibodies to effectively bind their target. While DNA probes capable of penetrating live cells have been developed (Veetil et al., 2020; Bucevičius, Gilat & Lukinavicius, 2020), their use is still fairly limited and generally confined within 2D monolayers.

The vital approaches for the quantification of cell viability and proliferation can be divided in three main classes: (i) chemical, (ii) optical and (iii) electrical.

Chemical assays (Fig. 2A) rely on the quantification of Lactate Dehydrogenase (LDH) in the culture’s supernatant (Lam et al., 2020; Vormann et al., 2018; Liaudanskaya et al., 2019; Liaudanskaya et al., 2020). This enzyme, present in all cells, is released in the media upon membrane damage and is thus an effective indicator of cell death (Kumar, Nagarajan & Uchil, 2018). For this reason, LDH quantification is particularly suited for the monitoring of cytotoxicity when testing new pharmacological treatments. Additionally, being a measurement conducted on the culture’s media, it eliminates any possible interference of the acquisition on cellular function and easily adapts to high throughput screenings and long term monitoring. On the flip side, it is an indirect method and it requires a non-negligible amount of cell death. As such, its scope of application is rather narrow and mainly focused on drug testing. The estimate of LDH concentration, furthermore, is an average of the whole culture that cannot capture behaviourally different microenvironments within the culture.

Figure 2 Non-destructive methods for viability quantification.

(A) Working principle of the LDH assay. (B) Representative image of a 3D cell culture obtained with light-sheet microscopy. Image reproduced from Alladin et al. (2020). (C) Example of 2D conductivity map reconstructed from EIT measurements. Image extracted from Wu et al. (2018).

Optical methods that preserve the culture’s structural integrity address this limitation (Li et al., 2016; Pan, Onda & Hirano, 2019; Christoffersson et al., 2019; Alladin et al., 2020). (Fig. 2B). They rely on advanced microscopy technologies (e.g., laser-based confocal, light-sheet) and fluorescent probes that preserve cell viability (Li et al., 2021c; Cheng et al., 2017) to produce a stack of images corresponding to different depths within the culture. This approach is characterized by high spatial resolution (∼100–500 nm Fouquet et al., 2015) and by the possibility of monitoring more than one variable at the same time. As an example, Liaudanskaya et al. (2020) combined the study of neural network destruction due to impact injury with the quantification of changes in collagen fibrillary structure, while Aguet et al. (2016) analysed how membrane remodeling changes throughout the cell cycle. The versatility and potential of these methods is, however, associated with lengthy and complex experimental procedures and with the need of acquiring highly specific, expensive instrumentation. In addition, the constraints on the maximum axial scan range of the microscope (i.e., the maximum depth that can be effectively imaged) make these techniques better suited for smaller SF cultures.

Electrical measurements represent an effective compromise. Indeed, while their resolution is not as high as that of the optical methods, they pose fewer limitations on culture size, and the instruments required for the measurement are not so expensive. This is partly due to the fact that impedance spectroscopy has been used for a number of years to monitor viability and growth in cell monolayers (Pérez et al., 2018; Voiculescu, Li & Nordin, 2020). The working principle behind this methods relies on the measurement of changes over time in electrical current, upon stimulation of the culture with the same sinusoidal voltage signal (Benson, Cramer & Galla, 2013; Elbrecht, Long & Hickman, 2016; Srinivasan et al., 2015). Cells are generally grown on conductive electrodes and their well defined electrical model, that combines resistant and capacitive elements, is used to infer the status of the culture. As such, the major issue when translating this approach in 3D is the lack of direct contact between the electrode and the cells. This is commonly addressed by including the whole culture (e.g., the scaffold or other extracellular components) in the measurement with the assumption of negligible changes in the electrical properties of the support structure (Pan et al., 2019; De León, Pupovac & McArthur, 2020; Pan et al., 2020). An alternative solution has been proposed by Inal et al. (2017), where a SB culture featuring a conductive polymer scaffold is presented.

Beside impedance spectroscopy, which provides a population-wise quantification of cell viability, electrical impedance tomography (EIT) has recently been proposed (Wu et al., 2018; Wu et al., 2019; Yang et al., 2019). This technique has been extensively used for the non-invasive visualization of internal structures of the human body (Shiraz et al., 2019; Murphy et al., 2017; Samoré et al., 2017), as it provides a map of the conductivity of one or more sections of the object of interest (Fig. 2C). While this resolution is not yet available for 3D cell cultures, where a uniform change in electrical properties is generally recovered, EIT holds great potential for the non-destructive monitoring of viability because changes in the setup (e.g., increasing the number of electrodes) and/or a more advanced reconstruction algorithm are expected to resolve this issue.

Several non-destructive approaches are already available for the monitoring of cell viability and proliferation. However, no method has emerged as overall better than the others and a case by case evaluation taking into account culture type and study aim is generally recommended.

Gene Expression

Gene expression profiles are an established approach for the quantitative evaluation of cell behaviour and how it changes upon external stimulation (Picone et al., 2020; Ciardulli et al., 2020) or when environmental conditions are modified (Gamerith et al., 2017; Brady et al., 2020; Lewis, Green & Shah, 2018).

As for the evaluation of cell viability, the most common approaches for the quantification of specific proteins or RNA sequences is associated with the destruction of the culture. Indeed a prerequisite of techniques such as micro-arrays, RT-PCR or RNA-sequencing is the lysis of the cells, while flow cytometry requires culture disgregation and immunohistochemistry/immunofluorescence, its fixation and slicing (Table 3). High throughput and transcriptome-wide technologies, are also widely used, but have similar requirements (Waylen et al., 2020).

Table 3 Gene expression assays.

Assay	References	
Microarray	Gamerith et al. (2017), Brady et al. (2020)	
RT-PCR	Henriksson, Gatenholm & Hägg (2017), Zhou et al. (2017), Lewis, Green & Shah (2018)	
	He et al. (2020), Henrionnet et al. (2020), Brady et al. (2020)	
	Devall et al. (2020), Schneider et al. (2020), Ciardulli et al. (2020)	
RNA sequencing	Devall et al. (2020)	
Flow cytometry	Zhou et al. (2017)	
Immunohistochemistry/ immunofluorescence	Zhou et al. (2017), Henrionnet et al. (2020), Brady et al. (2020)	
	Ciardulli et al. (2020), Schneider et al. (2020)	

The scarcity of RNA and proteins is often responsible for the need to have direct access to the cells or their content. Fluorescent probes able to enter the cells and bind to specific RNA sequences have been shown to provide sufficiently strong signals which can also be measured in entire organisms (Okamoto, 2019; Braselmann et al., 2018; Suseela et al., 2018; He et al., 2020). This feature suggests the potential transferability of this approach to 3D cell cultures even though specificity, accuracy and uptake efficiency can still be challenging and depend on the specific target.

An alternative strategy, that allows for the quantification of specific proteins, relies on genome editing techniques to couple the production of a fluorescent reporter to the expression level of a gene of interest (Koch et al., 2018; Di Blasi et al., 2021; Ceroni & Ellis, 2018). This approach is commonly employed in the synthetic biology field (Cortesi et al., 2017; Bandiera et al., 2016; Xu & Qi, 2019) and is characterized by high precision and accuracy. Modifying the genome of eukaryote/human cells is however an additional procedural step, which might not be feasible, or result in side interactions and unexpected behavioural changes.

Both vital fluorescent probes and gene editing allow for the possibility of monitoring gene expression changes over time and, preserving the distribution of cells within the culture, they allow to study how different microenvironments within the culture affect cell behaviour. They are however affected by the same limitations discussed in the previous section for optical methods. Indeed, advanced microscopy set-ups (e.g., confocal microscope) are needed to properly visualize the internal culture’s regions, and limitations on the maximum axial scan range might preclude the use of this technique for larger SB systems.

In addition, the number of genes that can be monitored at the same time is fairly limited, both due to the maximum number of fluorophores that can be effectively distinguished (about five but depends on instrument and application; Kleeman et al., 2018) and the need to modify each gene independently when a gene editing procedure is required.

As such, additional improvements are needed to open to the non-destructive large-scale evaluation of gene expression. A possible strategy to increase the throughput of these assays with technologies already available, relies on the combination of different readout methods. This approach has already been proposed for the visualization of complex structures within 3D plant tissues (Ursache et al., 2018) (Fig. 3), but it could be adapted, using vital fluorescence- and colorimetric-based methods, to the study of gene expression in 3D cell cultures. While potentially useful to increase the number of markers that can be monitored at one time, this approach has limited scalability and is substantially affected by the same limitations detailed above. As such transcriptome/proteome-wide vital analyses require significant advances with respect to current technology.

Figure 3 Image exemplifying the combination of fluorescent and colorimetric readout methods.

Image extracted from Ursache et al. (2018). © 2017 The Authors The Plant Journal © 2017 John Wiley and Sons Ltd.

Disposition and Morphology

A key feature of 3D cell cultures, that sets them apart from 2D approaches is the presence of gradients within the structure (e.g., nutrients, oxygen, waste products, drug levels) which effectively create multiple microenvironments capable of influencing cell behaviour. Hence the study of cell disposition within the culture, and how it changes over time acquires great relevance.

Optical methods are naturally suited for this analysis as they provide direct visualization of the culture. Common approaches involve staining of different cellular components (e.g., nucleus, cytoplasm, specific proteins) and their imaging through different microscopy techniques (Table 4). These methods were initially developed for tissue slices and as such require the sectioning of the culture. They are, however, fairly straightforward and a wealth of resources are available for their optimization and execution. More advanced techniques, like light-sheet and two-photon microscopy have also been used to study cell disposition within the culture (Table 4). These methods afford more comprehensive results, as the whole structure can be taken into account, but are associated with higher costs and the need for specific instrumentation.

As described in the cell viability and proliferation section, viable fluorophores are available and can thus be used to monitor cell status and position within the culture. As an example, Wang et al. (2018) were able to evaluate cell migration over a period of 3 h. The use of this approach is however limited by the need for a microscope environmental chamber, to maintain standard culturing conditions, and by the detrimental effect that long term exposure to the excitation light can have on the fluorophore and the cells.

Computational modeling has been shown to be an alternative approach for the study of cell distribution within the culture (Table 4). Indeed, multiple set-ups are available for the simulation of cell migration and the study of how different variables (e.g., cell–cell, cell–matrix interactions, phenotype) influence it. Among these tools, the computational scaffold simulator (SALSA) that we recently developed (Cortesi et al., 2020b; Cortesi et al., 2021a) is particularly relevant, as it is programmable and hence adaptable to different setups.

Table 4 Cell disposition assays.

Assay	References	
Staining	Griffith & Swartz (2006), Trappmann et al. (2017), Liverani et al. (2019)	
	Shichi et al. (2019), Zoetemelk et al. (2019), Bassi et al. (2020)	
	Lamparelli et al. (2021), Pasini et al. (2021)	
Light-sheet/two-photon microscopy	Accardo et al. (2018), Cong et al. (2019), Pan, Onda & Hirano (2019)	
	Christoffersson et al. (2019)	
Computational models	Celià-Terrassa et al. (2018), Kuzmic et al. (2019)	
	Cortesi et al. (2020a), Kim et al. (2020)	

While computational models are not measurements techniques, validated mathematical models have been shown to be able to provide relevant insights on complex biological phenomena (Cortesi et al., 2019; Jin et al., 2017b; Imle et al., 2019), effectively optimize experimental conditions (Hyndman et al., 2020; Tajsoleiman et al., 2018; Cortesi et al., 2020b) and predict experimental results (Tripathi et al., 2020; Trac et al., 2019; Celià-Terrassa et al., 2018). As such, a more extensive integration of these techniques within the experimental studies is warranted and expected to improve the analysis of 3D cell cultures.

Cell morphology is another key aspect of culturing cells in 3D, as different environments have been shown to be associated with alternative cell shapes (Randles et al., 2020; Zhang et al., 2020; Miller, Hu & Barker, 2020) which, in turn, result in radical changes in behaviour (Leggett et al., 2021; Venturini et al., 2020; Esfahani & Knöll, 2020).

Imaging techniques are again widely used, as they provide qualitative information on cell shape and dimension while allowing for the quantification of indices like eccentricity or aspect ratio (Pasini et al., 2021; Cortesi et al., 2018; Costa-Almeida et al., 2019). Furthermore, high resolution techniques such as transmission electron microscopy, allow for the visualization of subcellular features and of how they change when cells are cultured in 3D (Remuzzi et al., 2020).

Raman imaging has also been shown to be able to retrieve cell shape both in isolated cells (Jin et al., 2017a) and from within a 3D culture (Baldock et al., 2019) (Fig. 4). This technique relies on the acquisition of Raman spectra at different points within the sample, according to a specific grid pattern. Integrating this information allows to effectively map the whole culture with a micrometric resolution. The need for specific instrumentation reduces the applicability of this method that however remains very promising, as it has been shown to be proposed also as a non-destructive technique (Kallepitis et al., 2017).

Figure 4 3D Raman map of a cell adhered to a scaffold.

Image extracted from Baldock et al. (2019).

Computational modeling has also been shown to be useful to study how cells move and change shape (Ruan & Murphy, 2019; Peng, Vermolen & Weihs, 2021; Van Liedekerke et al., 2020; Ziebert & Aranson, 2016). These tools provide an accessible alternative to investigate cell morphology that could be fundamental for the study of this aspect in the short term.

Comprehensively, the measurement of cell disposition and morphology with non-destructive techniques is possible but still affected by many limitations. While technology developments are expected to bridge this gap, computational models offer a low cost versatile alternative which could be used both to replicate experimental behaviours and to evaluate the effect of changing parameters and environmental conditions.

Matrix Interaction

Another aspect tightly connected with cells disposition within the culture and their migration abilities is their interaction between the cells and among the cells and the extracellular matrix (Yamada & Sixt, 2019). This is a key feature of in vivo biology, connected with changes in gene expression (Tajik et al., 2016) and behaviour (Liverani et al., 2019), whose study in the lab has been enabled by 3D cell cultures. As an example, in vivo-like cell alignment, induced by the surrogate ECM features, was shown to result in improved tissue regeneration and repair (Lu et al., 2021; Li et al., 2021a). While a comprehensive analysis of cell–cell and cell-environment interactions is beyond the scope of this review, the interested reader is referred to Delle Cave et al. (2021) and Bechtel et al. (2021) for more information.

Multiple techniques are available for the analysis of the interaction between the cells and their environment. Advanced microscopy setups (transmission/scanning electron microscopes) grant a resolution high enough to resolve specific features of the material and how they change due to cell activity (Ahmad et al., 2017; Hermenean et al., 2017; Dinescu et al., 2019; Lee et al., 2018). These assays are generally coupled with the analysis of the expression of key genetic markers (Ahmad et al., 2017; Jang et al., 2017; Ma et al., 2017; Dinescu et al., 2019; Lee et al., 2018), the evaluation of the activity of specific enzymes (e.g., Alkaline phosphatase) (Ahmad et al., 2017; Hermenean et al., 2017) or the staining of relevant compounds (Dinescu et al., 2019; Lee et al., 2018).

As detailed in the previous sections, these methods are largely connected with cell destructive processing. While the alternative approaches already described maintain their general validity, other methods specific for the evaluation of changes in matrix properties have been proposed for the study of artificial bone substitutes (Cortesi et al., 2021b; Lovecchio et al., 2022; Arunngam et al., 2018). They are based on either EIT or spectroscopic techniques and aim at quantifying different calcium-based components that are typical of cell-induced mineralization.

Raman spectroscopy was shown in Arunngam et al. (2018) to be able to measure hydroxyapatite in microspheres of gelatin hydrogel containing pre-osteoblasts. This data, compared with the corresponding cell density, allowed for the analysis of intra-culture variability.

Our works on the topic (Cortesi et al., 2021b; Lovecchio et al., 2022) rely on a different experimental model that grants better reproducibility and a higher level of control. Indeed, the amount of minerals produced by the cells is highly dependent on the type of cells and the experimental protocol employed. To overcome this issue, we used alginate-based scaffolds that can be polymerised in presence of defined amounts of calcium carbonate to produce phantoms with highly reproducible mineral content. This difference could be quantified with either a custom-made spectrometer (Lovecchio et al., 2022) or an EIT system (Cortesi et al., 2021b) (Fig. 5). The former is a miniaturized system that yields scaffold-wide measurements and could be potentially integrated within a bioreactor to enable the automatic monitoring of mineralization. The latter, on the other hand, is procedurally more complex and time consuming, but it retrieves a bidimensional conductivity map of a section of the scaffold, which might potentially enable sub-scaffold resolution.

Figure 5 Schematic representation of the phantoms and results presented in Cortesi et al. (2021b), Lovecchio et al. (2022).

Measuring mineralization in artificial bone substitutes is at the forefront of the study of the interaction between the cells and their environment, due to the prominent changes in matrix composition and properties caused by the cells, but this is not the only situation in which cells have been shown to remodel their environment (Liverani et al., 2017). As such further analyses, and other technologies, are warranted to enable the monitoring of matrix properties in different setups and applications.

Conclusion

3D cell cultures hold great potential for the improvement of in vitro experimentation. Indeed they enable the study of the interaction among the cells and between the cells and the environment in a controlled yet realistic setting. As such, these methods could be used to study complex biological phenomena and further our understanding of cell behaviours normally observed only in vivo, like 3D cell migration or matrix remodelling.

The establishment of these advanced cell culture methods as a standard could also contribute to the reduction and replacement of animal studies, as the improved accuracy of in vitro results could decrease the need for in vivo experimentation without compromising the validity of the analysis (Ingber, 2020). This would result in both a reduction of the drug development costs, as laboratory studies require less resources than animal ones, and also in an improvement of the efficacy of the whole drug development pipeline. Indeed, most of the therapeutic treatments that enter the testing process don’t get approved due to lack of effectiveness (Takebe, Imai & Ono, 2018; Hay et al., 2014) and thus more accurate in vitro models, could enable the early identification of ultimately unsuccessful compounds.

Beside de novo drug development, 3D cell culture models could also aid the definition of new therapeutic indications for drugs already approved. This approach, often called drug repurposing, is more cost effective, as the safety of the compound has already been proven, and its large scale production is already available. However, it is affected by the same issues discussed previously (i.e., high withdrawal rate due to ineffectiveness). As such, the use of 3D cell culture models in the initial phases of screening could be fundamental to further improve the effectiveness of this approach.

Another key advantage of 3D in vitro cultures is the possibility of creating patient-specific models that, yielding information on each subject’s response to different treatments/stimuli, represents an important resource for personalized medicine. This novel treatment paradigm promises increased response and better prognosis, by taking subject-specific features into account when defining the therapeutic protocol (Mathur & Sutton, 2017; Goetz & Schork, 2018).

Accurate and realistic experimental models are, however, only part of the equation. Indeed the techniques used to quantify relevant features are also key, as they determine the usability of each system and its scope of application. The methods presented in this review are an important step in this regard, showing how multiple key features can be evaluated non-destructively. However, not all the proposed techniques have the same relevance, as some allow to measure more than one variable, potentially at the same time (Fig. 6). Optical and spectroscopic techniques are the most convenient, spanning three variables each, while chemical measurements quantifies only one. In Fig. 6 computational modelling is also associate to the evaluation of a single variable. This is not entirely true, as specialized models can be used to simulate all the quantities considered within this work (Cortesi et al., 2020b; Cortesi et al., 2021a; Cortesi et al., 2020a; Kim et al., 2018b; Sun & Hu, 2018; M Cortesi & E Giorda, 2022, unpublished data), but this analysis hasn’t been included, as measurement techniques were preferred whenever available.

Figure 6 Schematic representation of the techniques presented within this work (coloured ovals) and of their use to quantify different biological variables (text fields).

Comprehensively, the proposed methods open to the non-destructive monitoring of 3D cell cultures dynamic behaviour and thus offer the unprecedented opportunity of characterizing complex biological phenomena in controlled yet realistic conditions.

The authors are grateful to Joseph Lovecchio, PhD. for helpful feedbacks and discussion.

Additional Information and Declarations

Competing Interests

Author Contributions

Data Availability

The authors declare there are no competing interests.

Marilisa Cortesi conceived and designed the study, conducted the literature research, prepared figures and/or tables, authored or reviewed drafts of the paper and approved the final draft.

Emanuele Giordano conceived and designed the study, authored and reviewed drafts of the paper and approved the final draft.

The following information was supplied regarding data availability:

This is a literature review, no raw data/code were produced.

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
