# Peer review of "Non-destructive monitoring of 3D cell cultures: new technologies and applications"

_PeerJ, doi:10.7717/peerj.13338_

## Round 0.1 · original submission · Major Revisions

Dear Drs. Cortesi and Giordano:

Thanks for submitting your manuscript to PeerJ. I have now received three independent reviews of your work, and as you will see, the reviewers raised some concerns about the research. Despite this, these reviewers are optimistic about your work and the potential impact it will lend to research on methods for visualizing 3D cell cultures. Thus, I encourage you to revise your manuscript, accordingly, taking into account all of the concerns raised by the reviewers.

While the concerns of the reviewers are relatively minor, this is a major revision to ensure that the original reviewers have a chance to evaluate your responses to their concerns.

Please also ensure that your figures and tables contain all of the information that is necessary to support your findings and observations. The Methods should be clear, concise and repeatable. Please ensure this. Please address all of the typos. Missing (also outdated) references have been raised by the reviewers.

I look forward to seeing your revision, and thanks again for submitting your work to PeerJ.

Good luck with your revision,

-joe

·

Basic reporting

-In introduction section, limitations related to conventional 2D culture systems were missed. Please add relevant information.
-In Figure 1 legend, the term SB should be defined.
-For scaffold free 3D culture authors can refer to previously published articles as follows; https://doi.org/10.1186/s12645-020-00074-4
-The limitations to SF cellular aggregation were missed in this review article.
-There are some typos errors within the manuscript. Please consider the issues
-Authors are asked to debate about SB culture systems such as spherical hydrogels such as microspheres used in different experiments.
-The role of cell alignment, orientation and cell-to-cell interaction within 3D SB cultures should be debated in detail.
-The effect of micro and nanopattering in 2D and 3D culture system should be also considered.

Experimental design

-

Validity of the findings

-

Additional comments

-

·

Basic reporting

In the current review, Marilisa Cortesi and Emanuele Giordano summarize and compared the different methods for 3D cell cultures, which is very promising methodology to mimic the in vivo micro-environment during in vitro study. Especially, when the 3D fell culture were combined with high-through out drug screening and live Real-Time visualization. Therefore, I agree and favor this review very much. Although the 3D culture was initialed from 1980s, this technology was really restricted to popular utilization due to technical issues.

Experimental design

The background literature and study rationale clearly articulated.

Validity of the findings

No comment

·

Basic reporting

In the manuscript by Cortesi et al., the authors summarized the recent advances in the field of non-destructive monitoring of 3D cell cultures. The authors first gave a brief introduction to the 3D cell culture model, and then systematically reviewed the techniques to monitor the 3D cell culturing and the cell-matrix interactions. The reviewer has some minor comments:

1. The pink color of the matrix in Fig 1a and Fig 1c is similar to that of the cells. It is suggested to change the matrix color to blue for better presentation.
2. The DNA content-based techniques for cell viability evaluation are missing, such as Picogreen assay. The most common technique to determine cell proliferation, BrdU assay, is missing.
3. Fig. 2 should summarize all the techniques for the evaluation of cell viability and proliferation, not only showing the EIT technique.
4. There are too many simple lists of the techniques throughout this review. The authors should add more comparison and their opinion on these techniques. It is suggested to change the summary paragraphs to tables for easier reading, for example, Line 200-208. Page 5.
5. The citations are too old. Most of the citations are published three years ago. Although it is important to give the readers an outline of technique development, it’s better to add more recent publications to give up-to-date information.

Experimental design

Nil

Validity of the findings

Nil

Additional comments

Nil

---

## Round 0.2 · accepted · Accept

Dear Drs. Cortesi and Giordano:

Thanks for revising your manuscript based on the concerns raised by the reviewers. I now believe that your manuscript is suitable for publication. Congratulations! I look forward to seeing this work in print, and I anticipate it being an important resource for research on methods for visualizing 3D cell cultures. Thanks again for choosing PeerJ to publish such important work.

Best,

-joe

·

Basic reporting

Good

Experimental design

not applicable

Validity of the findings

inteperations are valid

Additional comments

-

·

Basic reporting

The authors have made all the appropriate changes in this revision. This manuscript could be accepted in its current form.

Experimental design

Nil

Validity of the findings

Nil

Additional comments

Nil